# Plant-Associated Bacteria as Sources for the Development of Bioherbicides

**DOI:** 10.3390/plants11233404

**Published:** 2022-12-06

**Authors:** Wei Fang, Fang Liu, Zhaoyuan Wu, Zhigang Zhang, Kaimei Wang

**Affiliations:** 1Hubei Biopesticide Engineering Research Centre, Hubei Academy of Agricultural Sciences, Wuhan 430064, China; 2National Biopesticide Engineering Research Centre, Wuhan 430064, China; 3Key Laboratory of Microbial Pesticides, Ministry of Agriculture and Rural Affairs, Wuhan 430064, China

**Keywords:** plant-associated bacteria (PAB), phytopathogenic bacteria, deleterious rhizobacteria (DRB), allelopathic bacteria, endophytic bacteria, plant-growth-promoting rhizobacteria (PGPR), bioherbicide, mode of action

## Abstract

Weeds cause significant yield losses in crop production and influence the health of animals and humans, with some exotic weeds even leading to ecological crises. Weed control mainly relies on the application of chemical herbicides, but their adverse influences on the environment and food safety are a significant concern. Much effort has been put into using microbes as bioherbicides for weed control. As plant-associated bacteria (PAB), they are widely present in the rhizophere, inside crops or weeds, or as pathogens of weeds. Many species of PAB inhibit the seed germination and growth of weeds through the production of phytotoxic metabolites, auxins, hydrogen cyanide, etc. The performance of PAB herbicides is influenced by environmental factors, formulation type, surfactants, additives, application methods, and cropping measures, etc. These factors might explain the inconsistencies between field performance and in vitro screening results, but this remains to be clarified. Successful bioherbicides must be specific to the target weeds or the coinciding weeds. Detailed studies, regarding factors such as the formulation, application techniques, and combination with cultivation measures, should be carried out to maximize the performance of PAB-based bioherbicides.

## 1. Introduction

Weeds influence crops directly or indirectly by competing with them for nutrition, water, and light. Some weeds suppress the growth of crops through allelopathy [1]. Many species of weeds also act as alternate or alternative hosts of insect pests and/or pathogens [2]. Weeds are also of concern regarding the health of humans and animals. Some weeds produce toxic alkaloids that are harmful to animals and humans. Weeds such as ragweed, mugwort, pellitory, chenopod, Russian thistle, plantain, and annual mercury produce clinically relevant weed-pollen allergens that represent a substantial portion of seasonal allergens [3]. Besides the impacts of weeds on health, it is estimated that the crop-yield losses caused by weeds account for over 30% of the total yield in some major crops [4]. Gharde et al. assessed yield and economic losses in 10 major crops in India, estimating a total economic loss of about USD 11 billion due to weeds alone [5]. Weed interference in soybean in North America from 2007 to 2013 caused an average of 52.1% yield loss, representing about USD 17.2 billion annually [6]; in corn, it caused 52% yield loss, which was equal to USD 28 billion annually [7]. It was estimated that weeds in rangeland cause losses of USD 2 billion annually in the USA through reducing the yield and quality of forage, interfering with grazing, poisoning animals, increasing the costs of managing and producing livestock, and reducing land value [8]. The average yield loss of dry beans due to uncontrolled weeds is 71.4% in North America, representing an economic loss of USD 722 m annually [9].

The use of cultivation techniques, mechanical operations, and chemical herbicides can reduce the yield losses caused by weeds. Weed control was a labor-intensive process before the invention of chemical herbicides. Although the discovery of chemical herbicides liberated the hands of farmers and significantly reduced yield loss caused by weeds, their adverse effects, such as the generation of residues in the environment and agricultural products and the development of resistance, have been well documented [10]. A number of weeds have developed resistance to chemical herbicides due to long-term and inadequate application. Currently, 267 species (154 dicots and 113 monocots) of weeds have been reported to be resistant to 21 of the 31 known herbicide sites of action and to 165 different herbicides [11]. It is urgent to develop safer alternatives for weed control [12]. Biological weed control has several advantages over other weed-control methods, including its positive effect on ecosystems and human health through reducing the herbicide residues in the soil, water, and agricultural products. Much effort has been put into using microbes as bioherbicides for weed control. These microbes include bacteria, fungi, and viruses, which might be weed pathogens or other plant-associated microbes [13,14]. Biological weed-control agents offer different modes of action for killing or inhibiting the growth of weeds, which makes them a solution for the control of herbicide-resistant weeds. Compared to chemical herbicides, the development of bioherbicides takes less time and is cheaper. The process of developing bioherbicides with microbes is shown in Figure 1.

Plant-associated bacteria (PAB), including bacteria in the soil adjacent to the rhizosphere, rhizobacteria, endophytic bacteria, and phytopathogenic bacteria, co-evolve with crops and weeds. Many PAB show suppressive, herbicidal, or phytotoxic activity against weeds. The application of PAB displaying herbicidal activity leads to the colonization of target weeds by bacteria and the production of phytotoxins in weeds or the adjacent environment. Such herbicidal PAB have a high potential for use in the development of bioherbicides. Some PAB, such as *Xanthomonas campestris* pv. *poae* JT-P482, *Pseudomonas fluorescens* D7, and P. *fluorescens* ACK55, have been used in the development of bioherbicides and applied for the control of annual bluegrass and brome, respectively [15,16,17].

In this review, we summarized the progress in bioprospecting PAB for their herbicidal activities and developing bioherbicides with PAB as the active ingredients, with a focus on screening strategies, the diversity of herbicidal PAB, and their modes of action. We also discussed the factors affecting the performance of herbicidal PAB and the inconsistency between the activity levels determined by in vitro screening and field efficacy. Additionally, we predict the future of bioherbicides originating from PAB.

## 2. Rapid Screening for Plant-Associated Bacteria with Herbicidal Activity

Considering the diversity of microbes associated with weeds and crops, the screening for herbicidal or weed-suppressive bacteria among PAB must be rapid, robust, and cost-effective. Unlike chemical herbicides, which have a single mode of action, herbicidal PAB function through multiple modes of action. High-throughput screening with target enzymes or receptors is not suitable for herbicidal bacteria. At present, screening for herbicidal PAB is normally based on the seeds, seedlings, or tissue cultures of weeds.

### 2.1. Screening through Calluses and Cell Cultures of Weeds

As many herbicidal bacteria act through the production of phytotoxic metabolites, it is possible to use the calluses or cell cultures of weeds to screen for phytotoxic PAB. A rapid screening method was established with cell cultures and callus tissues of leafy spurge (*Euphorbia esula*) for the screening of phytotoxic bacteria against *Euphorbia* sp. [18,19]. The activity of the bacteria was evaluated with the suspended cells of leafy spurge, and cell viability was assessed using an Evan’s blue bioassay. For screening with callus tissues of leafy spurge, ~500 mg fresh callus tissue was surface-inoculated with 10 μL of 10^8^ CFU/mL bacterial suspension in a petri dish. To increase the throughput of the screening, a 24-well microplate screening system was set up with the callus tissues of leafy spurge [20]. The callus tissues were placed onto a growth medium in 24-well plates, and each callus piece was inoculated with the bacterial cultures produced after 48 h fermentation. The effects of the bacteria were assessed according to the damage they caused to the callus tissues after 48 h of incubation. Using this rapid assay method, about 30% of the isolates of rhizobacteria from *Euphorbia* sp. were found to be phytotoxic to the callus tissues of E. *esula*, and the phytotoxicity was highly correlated with the activity experimentally determined using leafy spurge seedlings [20,21].

### 2.2. Screening for through Seeds, Seedlings, and Weed Cuttings

The primary screening of the herbicidal activities of PAB can be achieved through the establishment of a screening procedure based on sensitive plants or weeds. Lettuce (*Lactuca sativa*) is often selected because of its high sensitivity to the non-specific phytotoxic secondary metabolites produced by herbicidal microorganisms. A seed-germination assay with L. *sativa* was carried out to verify the phytotoxicity of deleterious rhizobacteria (DRB) isolated from the roots of leafy spurge with 30 μL of bacterial suspension per seed pre-germinated on 1% water agar [22,23,24,25]. The screening of rhizobacteria based on the seed germination of L. *sativa* could also be carried out in a box with germination paper [26]. Root length and root injury were evaluated to determine the phytotoxicity of the isolated bacteria. A high ratio of bacteria isolated from the rhizosphere or the endorhiza of leafy spurge significantly inhibited root growth and caused necrotic lesions [24]. A more popular method involves the direct use of weed seeds to screen PAB for the inhibition of seed germination; this has been carried out using the seeds of *Orobanche* sp. [27], *Parthenium hysterophorus* [28], *Bromus tectorum* [29], *Avena fatua* [30], and *Chenopodium album* [31], etc. The seeds of all species of weeds can be used to screen for microbes producing the phytotoxic metabolites that inhibit seed germination. Such seed-germination assays can be conducted in petri dishes, tubes, or boxes via the water–agar incorporation of the cell-free filtrate from the bacterial cultures or using filter paper by the direct application of the cell-free filtrate onto the seeds. The germination rate of the weed seeds is used to determine the effects of the bacteria. The effects of the cell-free filtrates on the root growth and the fresh and dry weight of the weed roots and shoots can be evaluated after a longer duration of incubation. Kremer and Souissi conducted secondary screening with stem cuttings of leafy spurge to evaluate its lettuce-targeting DRB and found that about 30% of the isolates were highly detrimental, causing an injury rating over 3 [24]. In most cases, the seedlings of weeds are used to test the herbicidal effects of the bacteria obtained from a primary screening based on pot or field-plot assays.

### 2.3. Other Screening Methods

As the plant growth inhibition or bioherbicidal activity of PAB might be attributed to the high production of phytohormones, screening for the production of phytohormones by PAB is carried out before screening with weeds in many studies. Target hormones include indole-3-acetic acid and 5-aminolevulinic acid. The production of 5-aminolevulinic acid by PAB isolates can be quantified as follows: the culture supernatants are added to acetylacetone and sodium acetate buffer, boiled for 15 min, and then reacted with Ehrlich reagent at room temperature. The mixture is then measured at 556 nm with a spectrophotometer 20 min after the reaction [31,32]. The production of indole-3-acetic acid (IAA) by PAB can be visualized by reacting the culture supernatants with Salkowsky reagent. The reaction mixture is measured at 535 nm 20 min after the reaction for the quantification of IAA [33]. Considering the mechanisms of action of DRB in inhibiting the growth of weeds, screening for the production of active compounds can simplify or accelerate the discovery of herbicidal PAB. As hydrogen cyanide produced by many bacteria was related to their phytotoxicity to plant, some researchers used chemical method to visualize the production of cyanide by PAB through placing filter paper soaked with picrate/Na_2_CO_3_ solution, dried on the lid of the petri dish [22,34,35]. The color of the filter paper changes from yellow to light brown, brown, or reddish brown depending on the amount of cyanide produced.

## 3. Bacterial Pathogens in Weeds as Microbial Herbicides

Many bacteria cause leaf spots and bacterial wilt in weeds. The application of weed-pathogenic bacteria is a standard method in weed management. Many studies have evaluated the potential of such weed-pathogenic bacteria for application as bioherbicides (Table 1). The most commonly studied bacterial pathogens of weeds are *Xanthomonas* spp. and *Pseudomonas* spp.

### 3.1. Xanthomonas *sp.*

*Xanthomonas* sp. represents a large group of bacteria that cause diseases in more than 400 species of plant hosts [66]. Many species of weeds host *Xanthomonas* sp. Due to the high specificity of *Xanthomonas*, it is possible to apply weed-pathogenic *Xanthomonas* sp. as a bioherbicide. The most commonly studied species is X. *campestris*. Different pathovars of X. *campestris* show different levels of specificity against weeds. X. *campestris* pv. *poannua* (or X. *campestris* pv. *poae*) was isolated from diseased P. *annua* with symptoms of bacterial wilt [36,37,38]. In a pot assay, symptoms of wilt in P. *annua* appeared 7 to 10 days after the application of X. *campestris* culture; about 30% of the annual bluegrass wilted within 1 week, and 60–70% of the annual bluegrass plants were dead 3 weeks after the culture application [39]. X. *campestris* pv. *poae* caused the systemic wilt of annual bluegrass when applied after mowing [37,38]. Johnson used X. *campestris* pv. *poae* to control annual bluegrass in Bermuda grass [39], and the control effect of the strain MB218 reached 82% at 10^9^ cfu/mL after three applications. The host range of the bacterium was restricted to *Poa* sp. It is possible to use X. *campestris* pv. *poa* to control unwanted *Poa* sp. in crop fields or invasive *Poa* sp. in turf. The strains of X. *campestris* pv. *poae* from California and Michigan showed good control of P. *annua* ssp. *annua* and P. *annua* ssp. *reptans* in a growth chamber assay, but their effects in field tests decreased to 11 and 7%, respectively [40]. A Japanese group also evaluated X. *campestris* pv. *poae* for the control of P. *annua*. Nishino and Fujimori distinguished the isolates of X. *campestris* pv. *poae* collected in Japan into three groups [41]. Of the three groups of X. *campestris* pv. *poae* isolates, strain JT-P482 showed pathogenicity against annual bluegrass. The application of the JT-P482 culture to precut annual bluegrass caused over 75% weight loss due to plant wilting and death [42], while no symptoms were observed in other turfgrasses. The control effects of JT-P482 on annual bluegrass in greenhouse were correlated with the bacterial concentrations, and affected by the temperature [43,44]. The population of JT-P482 decreased to almost 0 in non-sterile sandy loam soil 2 weeks after application [45]. X. *campestris* pv. *poae* JT-P482 was developed by Japan Tobacco Inc. as a bioherbicide with the commercial name Camperico^®^ for the control of P. *annua*. Savage and Haygood also isolated many strains of X. *campestris* from asymptomatic plants, such as annual bluegrass, crabgrass, goosegrass, and other species of weeds collected in field and turf locations. Of these strains, MB245, 246, 249, 250, 253, and 260 were virulent against annual bluegrass, while MB251 targeted crabgrass [46]. Under cool and cold conditions, strains MB245 and 249 showed greater control of annual bluegrass than MB218. MB249 showed good control for many species of *Bromus*. Compared with MB218, turf grasses showed less susceptibility to MB245. With the reclassification of X. *campestris*, the xanthomonads that attack grasses were recategorized as X. *translucens*. Mitkowski found that X. *translucens* pv. *poae*, the casual agent of bacterial wilt in annual bluegrass, showed high specificity for P. *annua* and only weak pathogenicity against P. *annua* var. *reptens* and *Poa trivialis* cv. *sabre* [47,48]. The X. *campestris* pathovar PT1 isolated from P. *trivialis* (a common weed contaminant of creeping bentgrass seed stock) selectively controlled P. *trivialis*. The X. *campestris* PT1 treatment of turfgrass with transplanted P. *trivialis* plugs caused about 19.4% of the P. *trivialis* to wilt [49]. Several isolates of *Xanthomonas* sp. from diseased common cocklebur (X. *strumarium*) leaves with angular-shaped spotting symptoms on the leaf margins and central leaf areas killed common cocklebur seedlings [50]. Of these *Xanthomonas* sp. isolates, LVA987 caused extensive blight in treated common cocklebur leaves 7–9 days after inoculation and the death of treated plants 10–12 days after treatment. The isolate LVA987 showed high pathogenicity against C. *canadensis*, A. *artemisiifolia,* A. *trifida*, and other Asteraceae plants. LVA987 was further characterized as X. *campestris*. The treatment with the LVA987 culture at 10^9^ cfu/mL caused significant mortality and a reduction in growth in the rosette and bolting growth stages of C. *canadensis* [51]. Dang et al. analyzed the bacterial populations in diseased exotic club moss *Selaginella moellendorffii* through metagenome profiling and found that X. *translucens* was present in many diseased samples but not in the native club mosses [67].

### 3.2. Pseudomonas *sp.*

P. *syringae*, with a very wide spectra of hosts, was listed as No. 1 of the top 10 pathogenic bacteria in agriculture in a survey organized by Molecular Plant Pathology [68]. Many pathovars of P. *syringae* are of economic importance, causing bacterial speck in tomato (P. *syringae* pv. *tomato*), bleeding canker in horse-chestnut (P. *syringae* pv. *aesculi*), and bean halo blight (P. *syringae* pv. *phaseolicola*). Many pathovars also infect weeds or undesirable plants. P. *syringae* pv. *phaseolicola* infects kudzu (P. *lobata*). Younger kudzu seedlings treated with P. *syringae* pv. *phaseolicola* displayed water-soaked symtoms and disease severities were increased with multiple appliction, but the plants treated with P. *syringae* pv. *phaseolicola* showed a higher regrowth potential [52]. P. *syringae* pv. *tagetis* is a pathogen of many plants, especially composite plants. Certain isolates of P. *syringae* pv. *tagetis* were found to infect marigold, sunflower, Jerusalem artichoke, and ragweed through wound inoculation [53]. Common ragweed (A. *artemisiifolia*) was also found to be a host of P. *syringae* pv. *tagetis* [54]. A typical symptom of P. *syringae* pv. *tagetis* infection is apical chlorosis, often accompanied by stunting. Early field trials showed that P. *syringae* pv. *tagetis* isolated from Canada thistle (*Cirsium arvense*) is more effective against annual weeds, such as A. *artemisiifolia*, C. *canadensis*, *Lactuca serriola*, and X. *strumariaum* [55,56,57]. P. *syringae* pv. *tagetis* reduced the shoot growth of common ragweed and Canada thistle by 82 and 31%, respectively; the common ragweed treated with P. *syringae* pv. *tagetis* showed chlorosis in 60% of the leaf area [58]. Tichich and Doll applied the sap of Canada thistle naturally infected with P. *syringae* pv. *tagetis* to control Canada thistle in the field [59]. Although the control effect was not influenced by the sap concentration and the spray volume, the consecutive spraying of the sap increased the infection rate and enhanced the disease incidence. P. *syringae* pv. *tagetis* was also observed to infect *Ambrosia grayi*; almost 100% of the A. *grayi* became infected 6 weeks after a single dose of P. *syringae* pv. *tagetis,* with application weekly or every 3 weeks [60]. An uncharacterized pathovar of P. *syringae*-CT99B016C, with differences to P. *syringae* pv. *tagetis*, was found to be pathogenic to Canada thistle, causing moderate to severe apical chlorosis and similar symptoms in *Sonchus* (moderate chlorosis in new growth areas) [61]. The strain CT-99B106C also caused severe disease in S. *oleraceus* and S. *asper* (annual and spiny sowthistle) and T. *officinale* (dandelion). The metagenome profiling of diseased exotic club moss S. *moellendorffii* demonstrated the presence of P. *syringae* in many diseased moss samples [67]. P. *aeruginosa* strain CB-4 isolated from infected corn leaves produced a herbicidal compound that strongly inhibited the growth of D. *sanguinalis* [62].

### 3.3. Other Phytopathogenic Bacteria

Besides *Pseudomonas* spp. and *Xanthomonas* spp., other phytopathogenic bacteria have also been studied for weed management. The pathogen that causes ginger (*Zingiber officinale*) bacterial wilt, R. *solanacearum*, was investigated for the biocontrol of Kahili ginger (H. *gardnerianum*), an invasive weed in the tropical forests of Hawaii. The inoculation of R. *solanacearum* through stem injection or root cutting caused H. *gardnerianum* plants to exhibit irreversible chlorosis and severe wilting 3–4 weeks following inoculation. This inoculation also caused systemic infection in the rhizomes, which led to their death and decay [63]. Bacteria isolated from the diseased leaves of C. *canadensis* infected the healthy leaves of C. *canadensis* and inhibited the seed germination of rye grasses and *Cichorium intybus* by over 90% [69]. B. *andropogonis* isolated from diseased chickweed showed herbicidal activity against weeds in the Caryophllaceae, Poaceae, and Fabaceae families, but the infection of B. *andropogonis* required wounds and infiltration under natural conditions [64]. B. *cereus* XG1 isolated from watermelon showing symptoms of bacterial fruit rot exhibited herbicidal activity against D. *sanguinalis* [65].

## 4. Phytotoxic or Herbicidal Rhizobacteria

### 4.1. Deleterious Rhizobacteria (DRB) with Herbicidal or Phytotoxic Activities

Deleterious rhizobacteria (DRB) are predominantly saprophytic bacteria that aggressively colonize plant seeds, roots, and rhizospheres and readily metabolize organic substances released by plant tissues [70]. DRB exist widely in the rhizosphere of weeds and crops. Many studies have focused on screening DRB for the ability to inhibit or kill weeds while promoting or not affecting the growth of crops (Table 2). A high ratio of active DRB isolates was found among the rhizobacteria of economically important weeds, such as common cocklebur, common lambsquarters, jimsonweed, morning glory, Pennsylvania smartweed, redroot pigweed, and velvetleaf [71,72]. Host plants might mediate the establishment and colonization of DRB by the exudates of their seeds and seedlings [73,74].

Downy brome is a serious problem in wheat fields. P. *fluorescens* strain D7 isolated from the roots of winter wheat inhibited downy brome growth in laboratory bioassays [29] and under a controlled environment [75]. The cell-free filtrate of strain D7 strongly reduced the root growth and inhibited the seed germination and root growth of many species of *Bromus* [76], but was safe for other tested plants, especially wheat and dicotyledon plants. It significantly reduced the root mass of two varieties of B. *tectorum* by 42~64%. P. *syringae* strain 3366 from the rhizoplane of wheat and downy brome was active against downy brome in different assays [29]. Three strains of non-fluorescent pseudomonads isolated from the rhizosphere of downy brome strongly inhibited the root growth of downy brome, with reductions ranging from 64 to 80% in in vitro screening and 27 to 35% in a growth chamber assay [104], and one of the three non-fluorescent pseudomonads, strain NRRL-18295, reduced stand and shoot growth by 35 and 45%, respectively. P. *putida* FH160, E. *taylorae* FH650, and X. *maltophila* FH131 were isolated from the rhizoplane of downy brome and other plants [77,78]. FH160 and FH650 strongly inhibited the root growth of downy brome, Japanese brome, and jointed goatgrass, while FH131 only showed activity against downy brome and jointed goatgrass in agar plate assays. However, in the pot assays in the growth chamber, FH160 and FH131 only showed activity against downy brome, while FH650 only targeted jointed goatgrass through soil surface dripping. All three strains suppressed the growth of downy brome and jointed goatgrass, with reductions in biomass ranging from 43 to 77% in field plots. Kennedy and Stubbs found that about 3.1% of the isolates from the rhizosphere of winter wheat, jointed goatgrass, and downy brome inhibited jointed goatgrass but not winter wheat in the in vitro screening, and further screening via field studies confirmed the growth inhibition of jointed goatgrass by four strains [105]. P. *trivialis* X33d isolated from the rhizosphere of durum wheat (*Triticum durum*) seedlings strongly reduced the dry weight of the roots and shoots of B. *diandrus* and promoted the growth of many species of crops [79,80].

Many DRB also inhibit the growth of other monocotyledon weeds. P. *fluorescens* WH6 isolated from rhizosphere soil inhibited the germination of the seeds of grassy weeds, such as annual bluegrass (P. *annua*), without significantly affecting the growth of established grass seedlings and mature plants or the germination of the seeds of broadleaf plant species [81]. Some deleterious P. *fluorescens* strains reduced the populations of the target invasive annual grasses in invaded pastures and grassland to nearly zero after 5 years of application and maintained control for 7 years after application [106]. P. *fluorescens* strain BRG100, isolated from the rhizosphere of green foxtail (*Setaria viridis*), strongly inhibited the root elongation of green foxtail by 74% [82]. DRB from wheat, namely P. *rettgeri* strain CPS67 and *Pseudomonas* isolate HWM11, strongly suppressed the growth of P. *minor* in pot assays [83]. An unidentified DRB strain Pk2 isolated from the rhizosphere of P. *conjugatum* in a soybean field inhibited the germination of seeds and the growth of P. *conjugatum* [84]. DRB isolated from weeds associated with wheat significantly reduced the germination, dry matter, root length, and shoot dry weight of wild oat, little-seed canary grass, and broad-leaved dock, while some promoted the germination and growth of wheat [107]. P. *kilonensis*/*brassicacearum* strain G11, a cyanide producer isolated from *Galium mollugo,* strongly inhibited the growth of E. *crus*-*galli,* reducing the above-ground biomass by 95% and decreasing the root length [85]. The cell-free filtrate of *Chromobacterium* sp. S-4 isolated from the rhizosphere of D. *sanguinalis* reduced the fresh weight of D. *sanguinalis* by 60% and caused chlorosis in the treated leaves 3 days after application [86]. A consortium of three species of *Pseudomonas* sp. isolated from the rhizosphere of wheat caused up to 50.0% and 56.7% mortality in A. *fatua* and P. *minor* seedlings and reduced root length by up to 73.8% and 53.9%, respectively. Additionally, these consortium strains increased wheat shoot length, root length, fresh biomass, dry biomass, and leaf greenness by up to 41.6, 100, 79.9, 81.5, and 21.1%, respectively [35]. Li et al. isolated several herbicidal strains of rhizobacteria against wild oat, and one of the strains, *Bacillus* sp. X20, inhibited the germination of seeds of wild oat by 75% [30]. P. *fluorescens* strain L2-19, S. *maltophilia* strain TFR1, and P. *putida* strain B1-7 isolated from the rhizosphere of green foxtail significantly reduced the growth of green foxtail [87]. P. *fluorescens* G2-11 inhibited the germination of seeds and the root growth of S. *viridis* in silt loam and sandy loam [88]. Kennedy screened large quantities of bacteria from the soil and rhizoplane of plants in winter wheat fields and found that three P. *fluorescens* strains, XJ3, XS18, and LRS12, inhibited the growth of annual bluegrass (P. *annua*), with shoot inhibition of 42–78% and root inhibition of 39–79% over 2-year field tests [89]. Of the rhizobacteria and soil bacteria isolated from stunted weeds, about 45% suppressed the root growth of downy brome, jointed goatgrass, and/or medusahead, but only 18% did not suppress the growth of non-target plants in the root-length assays [106]. Further screening led to the discovery of five highly active isolates suppressing downy brome, jointed goatgrass, and medusahead without inhibiting the growth of four winter-hardy non-target plants. Of these five isolates, three belonging to P. *fluorescens* consistently inhibited the growth of the three weeds by 85 to 100% in the small-field studies. Field studies consistently showed a 50% reduction in downy brome, jointed goatgrass, and medusahead after three years of bacterial application. An almost complete suppression of these fall annual grass weeds was seen 5–7 years after a single application in fields containing desirable plants (pasture grass, winter wheat (*Triticum aestivum* L.), perennial bunchgrasses, and native grasses).

Leafy spurge is a serious problem in North America. Many DRB with phytotoxic activity have been isolated from the rhizosphere of leafy spurge [18]. P. *fluorescens* isolate LS102 and F. *balustinum* isolate LS105 inhibited the root growth of leafy spurge by 80 and 65%, respectively, in a hydroponic system [90]. P. *fluorescens* LS102 and LS174 reduced the root weight and root carbohydrate content of leafy spurge by 20% [91]. Many DRB from the rhizosphere of Palmer amaranth (A. *palmeri* S. Wats.) strongly inhibited the plant’s seed germination, with *Pseudomonas* sp. strain TR10 reducing the seed germination rate to 5.26% and the seedling length by 92.85% in in vitro assays [92]. The application of deleterious cyanogenic P. *fluorescens* and A. *delafieldii* reduced the fresh and dry shoot weight of velvetleaf, while they showed no effects on the growth of corn [93]. DRB from the rhizosphere of S. *acuta*, such as *Xanthomoas* sp, P. *aeruginosa*, P. *fluorescens*, B. *subtilis,* and B. *cereus*, significantly reduced the germination of seeds of S. *acuta* and inhibited the growth of roots [94]. The P. *aeruginosa* isolate KC1 from the rhizosphere of castor plants (R. *communis*) reduced the root and shoot length of seedlings of A. *spinosus* and P. *oleracea* without inhibitory effects on wheat [95]. The DRB strain A08 isolated from A. *conyzoides* in a soybean field strongly inhibited the germination of seeds and the growth of A. *conyzoides* [84]. P. *aeruginosa* FS15, isolated from the soil adjacent to *Chenopodium* sp., completely inhibited the seed germination and root growth of C. *arvensis* and P. *oleracea* [96]. Rhizobacteria from the rhizoplane of white clover and ryegrass, *Pseudomonas asplenii* and P. *syringae*, significantly reduced the seedling dry weight and nitrogen fixation of white clover [97]. The DRB P. *fluorescens* S61, showing the strong inhibition of *Rhizobium leguminosarum,* significantly inhibited the growth of white clover on a golf course [98]. The DRB strain E. *taylorae,* isolated from weeds producing a high amount of indole acetic acid, inhibited the root growth of several species of weeds, including bindweed, velvetleaf, pigweed, green foxtail, and morning glory [99]. The DRB P. *fluorescens* WSM3455 and WSM3456 and A. *xylosoxidans* WSM3457, isolated from weeds in a vineyard, specifically inhibited the growth of wild radish and caused some foliar symptoms, such as growth retardation, leaf chlorosis, and poor lateral root development [100]. P. *putida* T42, P. *fluorescens* L9, P. *fluorescens* 7O_0_, P. *aeruginosa* O_0_10, and P. *alcaligenes* W9, isolated from weed-infested wheat soil, significantly reduced the number of plants, root dry weight, seed production, and straw weight of broad-leaved dock, with maximum reductions of around 50%, while all the strains except strain O_0_10 increased the wheat yield by 58~86% compared with the weedy control [101]. Five uncharacterized DRB isolates consistently inhibited the seed germination and growth of weeds such as P. *conjugatum*, A. *conyzoides*, *Chromolaena odorata,* and A. *spinosus* [108]. For A. *conyzoides*, the inhibition of seed germination and growth by isolates BL03 and BL07 was over 90%. B. *flexus* JMM24 isolated from the rhizosphere of L. *aphaca*, a common weed in mustard fields, caused a significant growth reduction in L. *aphaca* of up to 92%, while it increased the growth of mustard by up to 191% in terms of root dry weight and 119% in terms of shoot dry weight in pot assays [32]. B. *japonicum* isolate GD3 and P. *putida* GD4 inhibited the root elongation of *Ipomoea hederacea* by about 26 and 90%, respectively, 7 days after inoculation [102].

Certain parasitic weeds, such as *Striga* sp. and *Orobanche* sp., infest many crops and cause significant losses in crop production. The DRB P. *fluorescens* strain QUBC3, isolated from the roots of *Phalaris* sp., strongly inhibited the radical growth of O. *aegyptiaca* and O. *cernua* [27]. Of the 460 isolates from soils naturally suppressive to S. *hermonthica*, 15 P. *fluorescens*/P. *putida* isolates significantly inhibited the germination of S. *hermonthica* seeds and significantly reduced the emergence of shoots from treated maize [103]. Several isolates isolated from S. *hermonthica*-suppressive soil, belonging to the *Bacillus*, *Streptomyces,* and *Rhizobium* genera, caused seed decay in S. *hermonthica* [109].

### 4.2. Plant-Growth-Promoting Rhizobacteria (PGPR) with Herbicidal or Phytotoxic Activities

Some PGPR also show herbicidal activity against weeds. Bacteria associated with faba bean (*Vicia faba*), the host of *Orobanche* sp., inhibited the seed germination of *Orobanche* sp. and the emergence of shoots. P. *fluorescens* strain Bf7-9, isolated from the rhizosphere of faba bean, positively influenced the growth of faba bean but reduced the pre-emergence of *Orobanche crenata* and *Orobanche foetida* by 71 and 81%, respectively, in root chamber assays. Soil drenching with Bf7-9 significantly reduced the number and dry weight of emerging shoots in O. *crenata* and O. *foetida* by 64 and 76% and 39 and 63%, respectively [110]. *Pseudonomuas marginalis* strain Nc1-2 from the rhizosphere of faba bean promoted the growth of faba bean; meanwhile, it reduced the number emerging shoots in O. *crenata* [110]. The culture of another isolate of P. *fluorescens* also reduced the number of emerging shoots in O. *crenata* by 75% 2 months after soil treatment [110]. B. *subtilis* and *Bacillus pumilus* reduced the number of emerging shoots and the fresh and dry weight of O. *crenata* 4 months after soil treatment [111]. *Bacillus circulans* and *Bacillus megaterium* var. *phosphaticum,* isolated from the soil of a faba bean field, strongly reduced the germination and growth of O. *crenata* [112]. The combination of B. *megaterium* var. *phosphaticum* with *Rhizobium leguminosarum p*v. *viceae* significantly reduced the seed germination of O. *crenata*. Faba bean bacterized with the combination of B. *megaterium* var. *phosphaticum* and *Rhizobium leguminosarum p*v. *viceae* reduced significantly the shoot emergence and dry weight of O. *crenata* in a field test [112]. *Bacillus velezensis* JTB8-2, isolated from the rhizosphere soil of Egyptian O. *aegyptiaca,* promoted the growth and yield of tomato at a high application rate (3.2 L/105 m^2^), while it significantly reduced the number of emerging shoots, fresh weight, and biomass of O. *aegyptiaca* in pot and field experiments and the parasitic rate in the field [113].

The plant-growth-promoting bacterium B. *subtilis* strain SYB101 isolated from the wheat rhizosphere caused 70.8 and 80.7% decreases in the root and shoot dry weight of P. *minor*, respectively, in pot assays [82]. *Bacillus siamensis* RWA52 and *Bacillus endophyticus* RWA69 isolated from rhizosphere of wheat caused a significant increase in the dry weight of the roots and shoots of wheat and decreased the dry weight of the shoots and roots of A. *fatua* [114]. Multiple spraying applications of P. *fluorescens* significantly reduced the invasiveness of P. *annua* in turfgrass football pitches by reducing the number of tillers [115].

### 4.3. Allelopathic Rhizobacteria as Biohercides

Allelopathy was defined by Rice [116] as “the interaction of one plant or microorganism on another, whether stimulating or inhibiting, through the release of chemical compounds which pass through the environment”. Some rhizobacteria can proliferate and release phytotoxic compounds in the rhizosphere, which can be adsorbed by weeds and cause growth retardation or other phytotoxic symptoms on weeds. The phytotoxicity of such rhizobacteria has high host specificity. Kremer termed such rhizobacteria “allelopathic bacteria” [19]. There are two types of allelopathic rhizobacteria—DRB and PGPR. We apply this term as it was used in the original studies mentioned above, without further separating the category into DRB or PGPR. P. *putida* strain NBRIC19 isolated from plants grown in the close vicinity of P. *hysterophorus* enhanced the growth of wheat in the presence of *Parthenium* [117]. Treatment with P. *putida* NBRIC19 reduced the inhibition of the growth of wheat by nearby *Parthenium*, which resulted in 52.29, 28.73, and 76.31% increases in root length, shoot length, and dry weight, respectively, as compared with the control. P. *putida* NBRIC19 increased the species diversity of microbes in a *Parthenium*-invaded area and controlled the growth of other weeds such as *Commelina benghalensis* and *Cynodon dactylon* [118]. Abbas et al. conducted the large-scale screening of allelopathic rhizobacteria from the rhizosphere of wheat and weeds in a wheat field and found that the dry matter of broad-leaved dock, wild oat, little-seed canary grass, and common lambsquarters were reduced by eight strains (23.1–68.1%), seven strains (38.5–80.2%), eight strains (16.5–69.4%), and three strains (27.5–50.0%), respectively [25]. Among these active allelopathic rhizobacteria, four strains ((L9, T42, 7O_0_, and O_0_10) suppressed little-seed canary grass, which improved wheat grain yield losses by 20.1 to 66.9% in pot assays and 34.3 to 64.3% in the field trial [119]. The four strains, L9, T42, 7O_0_, and O_0_10, were characterized as P. *fluorescens*, P. *putida*, P. *aeruginosa,* and P. *fluorescens,* respectively [120]. *Enterobacter* sp. strain I-3, isolated from the soil of an agricultural field, significantly inhibited the seed germination and growth of *Cyperus microiria*. The culture of strain I-3 also reduced the seed germination of D. *sanguinalis* [121].

## 5. Endophytic Bacteria with Phytotoxic Activities

Endophytic bacteria have been isolated from diverse plants and show beneficial effects on the growth of host plants and the tolerance of host plants to biotic and abiotic stress. However, some endophytic bacteria show phytotoxic activity or produce herbicidal secondary metabolites. The endophytic bacterium *Klebsiella pneumoniae* strain YNA12 from evening primrose strongly inhibited seed germination in evening primrose and reduced the seedling length and biomass [122]. *Pseudomonas viridiflava* strain CDTRc14, an endophyte isolated from *Lepidium draba* in a vineyard, inhibited the seed germination of L. *draba* and reduced the biomass and the root length of L. *draba* seedlings [123,124]. A bacterial endophyte *Pseudomonas brassicacearum* YC5480 from *Artemisia* sp. showed phytotoxicity to some monocotyl and dicotyl plants [125]. Several species of endophytic actinomycetes from different plants showed phytotoxicity against the seedlings of A. *conyzoides*, *Bidens biternata*, and P. *hysterophorus*, with symptoms such as leaf curling, wilting, and burning, while some reduced the root and shoot growth [126].

## 6. Other Plant-Associated Bacteria with Phytotoxic Activity

In this section, PAB that cannot be assigned to the above classifications are discussed. *Serratia plymuthica* A153 isolated from the roots of winter wheat selectively suppressed the growth of many species of dicotyledon weeds and crops but not the monocotyledon species of *Poaceae* [127]. In a field test using winter wheat, spring barley, and potato, S. *plymuthica* A153 suppressed a range of weeds, such as C. *album*, *Stellaria media*, *Polygonum convolvulus,* and *Galeopsis speciosa,* etc. [128], but this suppression was only observed shortly after spray application, while fast regrowth and no significant effects were seen two months after application. *Azospirillum brasilense* L2 and L4, nitrogen-fixing bacteria isolated from the soil of a sorghum field, completely inhibited the seed germination of S. *hermonthica*, while strain L4 promoted the growth of sorghum [129]. The cell-free supernatant of the soil bacterium X. *campestris* pv. *retroflexus,* isolated from redroot pigweed, strongly inhibited the growth of *Capsella bursa*-*pastoris*, P. *oleracea,* and *Amaranthus retroflexus*, even at 30-times dilution, with reduction in shoot length and root length ranging from 52.6 to 74.8% and 72.1 to 83.6%, respectively [130].

## 7. The Modes of Action of the Herbicidal Activity of Plant-Associated Bacteria

The possible modes of action of herbicidal PAB include the following mechanisms: the production of phytotoxic metabolites, the production of exopolysaccharides, the overproduction of auxins, and other unclear modes.

### 7.1. Production of Phytotoxic or Herbicidal Metabolites

The phytotoxicity of weed bacterial pathogens might be caused by the production of phytotoxins. P. *syringae* pv. *tagetis* produces tagetitoxins, which disrupt the biogenesis of chloroplasts, resulting in the chlorosis of newly developed leaves in treated plants [57]. Many DRB induce damage to weeds through the production of phytotoxic metabolites or phytotoxins. Several DRB produce germination-arrest factors that inhibit the germination of weed seeds in a developmentally specific manner, typically irreversibly blocking the germination process immediately after the emergence of the plumule and coleorhiza [131,132]. P. *fluorescens* WH6 produced 4-formylamino- oxyvinylglycine (FVG), an herbicidal germination-arrest factor (GAF) that irreversibly inhibited the seed germination of grassy weeds, such as annual bluegrass, without significantly affecting the growth of established seedlings and mature plants of annual bluegrass or the seed germination of broadleaf plant species [81,133,134]. FVG was also found in other strains of P. *fluorescens*, namely *Pseudomonas synxantha*, *Pseudomonas chlororaphis*, and *Pantoea ananatis* [135]. The inhibition of root elongation in downy brome by P. *fluorescens* strain D7 was related to the production of a phytotoxic metabolite; at a concentration of 8%, the cell-free filtrate of strain D7 reduced the root elongation of downy brome by about 50% [136], and the phytotoxins inhibited the root and shoot growth of downy brome by 80% in a seed germination assay at 2 and 400 ng/m, respectively [137]. The active fraction further isolated from the culture of D7 completely inhibited the growth of downy brome at 1 mg/L [138]. The herbicidal activity of P. *aeruginosa* H6 was attributed to the production of quinoline and its derivatives [139]. Pseudophomin A, produced by P. *fluorescens* BRG100, strongly inhibited the root elongation of green foxtail by 67% [140,141]. The diketopiperazines produced by B. *velezensis* JTB8-2 inhibited the seed germination of O. *aegyptiaca* at 0.5~4.0 mM [114]. The phytotoxins produced by X. *campestris* pv. *retroflexus* that contributed to its toxicity to A. *retroflexus* were cyclo-(Pro-Phe) and six organic acids [142], which first made the thylakoids of chloroplasts and the cristae of mitochondria swell, before finally disrupting the cell membranes, causing the chloroplasts to disintegrate and the mitochondria to vesiculate [143]. 2-Hydroxymethyl-phenol, produced by P. *aeruginosa* C1501, caused a large area of necrosis on the leaves of *Amaranthus hybridus* but was safe for sorghum [144]. The phytotoxins produced by P. *syringae* strain 3366 controlled corn spurrey (*Spergula arvensis* L.) and fireweed (*Epilobium angustifolium* L.) seedlings at 10 ppm and reduced the root and shoot growth of cuttings of the perennial weeds birdsfoot trefoil (*Lotus corniculatus* L.) and silverleaf (*Potentilla pacifica* Howell) at 103 ppm [145]. The phytotoxic compounds produced by P. *syrinagae* strain 3366 included phenazine-1-carboxylic acid (PCA), 2-amino phenoxazone, and 2-amino phenol, of which PCA was the major phytotoxic metabolite. PCA inhibited downy brome root growth by 99% at 5.7 mg L^−1^ [146]. These phytotoxic compounds were also found to be produced by strain 3366 in field soil. The inhibition of the shoot growth of morning glory by P. *putida* GD4 was attributed to its production of phytotoxins [108]. The culture filtrate extract of P. *aeruginosa* CB-4 strongly inhibited radicula and coleoptile elongation [65]. Leaf necrosis in A. *hybridus* and E. *crus*-*galli* caused by the cell-free filtrate of P. *aeruginosa* strain B2 isolated from the rhizosphere of wheat indicated the production of phytotoxic metabolites by the strain, and the isolated active compound completely inhibited the seed germination and seedling growth of A. *hybridus* and E. *crus*-*galli* at concentrations of 10 and 15 mg/L, respectively [147].

Lipopeptides from *Bacillus clausii* isolated from *Dracocephalum tanguticum* showed herbicidal activity against P. *annua* at 32 ppm [148]. The growth retardation in L. *aphaca* caused by the deleterious rhizobacteria of *Brassica juncea* could be attributed to the production of 5-aminoleveulinic acid [32]. A strain of rhizobacterium, B. *cereus* 57-02, produced an unidentified phytotoxin, causing rot in the rootlets of lettuce seedlings at 57 mg/L [26].

### 7.2. Production of Hydrogen Cyanide

Cyanide has been associated with the phytotoxicity of many rhizobacteria against weeds. P. *fluorescens* S241 increased the cyanide content in treated soil to inhibit the growth of bean and lettuce [22]. The inhibition of the root growth of *Sida acuta* by rhizobacteria could be attributed to the production of cyanide [94]. The studies by Blom et al. demonstrated the function of the hydrogen cyanide produced by P. *aeruginosa* PAO1 in the volatile-mediated killing of *Arabidopsis thaliana* [149]. The diminished production of hydrogen cyanide by P. *aeruginosa* PAO1 resulted in a strong reduction in volatile-mediated phytotoxicity. The deleterious pseudomonads from the rhizosphere of weeds in an alfalfa field also produced hydrogen cyanide, which could explain its inhibition of the root growth of the weeds *Axonopus affins* (Chase) and *Lens esculenta* (Moench) [150].

### 7.3. Production of Exopolysaccharides

Some rhizobacteria isolated form *Euphorbia* spp. produced exopolysaccharides that significantly reduced the growth of leafy spurge calluses [151]. Although no direct evidence was found for the weed suppressive activity of exopolysaccharides, a relationship was observed between the production of exopolysaccharides and the growth of leafy spurge calluses.

### 7.4. Overproduction of Auxins

An appropriate concentration of auxins promotes the growth of plants, while a high concentration of auxins inhibits plant growth. PAB suppressed the growth of weeds through the high production of auxins, especially indole-3-acetic acid (IAA). E. *taylorae* produced a high amount of auxins (72 mg/L IAA) and inhibited the root growth of field bindweed (C. *arvensis*) by 90.5% [99]. The inhibition of the root elongation of morning glory by B. *japonicum* might be related to its high production of IAA [101]. The combination of *Enterobacter* sp. I-3 and the precursor of IAA significantly decreased leaf length, leaf width, and root length and increased the number of lateral roots in lettuce assays [152]. The inhibition of seed germination and seedling growth in evening primrose by *Klebsiella pneumoniae* YNA12 was partially attributed to its high production of IAA [121]. The high production of IAA might also be related to the inhibition of the growth of durum wheat by some *Pseudomonas* sp. [33].

### 7.5. Other Mechanisms in Plant Growth Inhibition

*Curtobacterium* sp. MA01 suppressed the growth of petunia plants by causing oxidative stress and decreasing the amount of antioxidants. *Curtobacterium* sp. MA01 also decreased protein synthesis in the infected petunia plants [153]. *Enterobacter* sp. I-3 controlled plant growth by inhibiting the gibberellin biosynthetic pathway and regulating abscisic acid and amino acid synthesis [154]. Certain rhizobacteria inhibited the growth of weeds through the production of volatile ammonia [155]. P. *putida* NRBIC19 reduced the allelopathy between P. *hysterophorus* and neighboring plants by altering the soil microbial diversity nearby [117]. The *SucB* gene in strain NBRIC19 participated in the degradation of the toxic allelochemicals produced by P. *hysterophorus* [156]. The suppressive activities of DRB against weeds might be related to their colonization on the roots of target weeds. The population of P. *putida* GD4 on the root surface of morning glory was about 100 times higher than that of the other investigated DRB, B. *japonicum* GD3, while both strains reduced the growth of morning glory, by 90 and 26%, respectively [102].

## 8. Limitations and Constraints to the Development of Bioherbicides with PAB

Up to now, very few PAB have been successfully developed as bioherbicides, but there are many reports of their unstable performance or even their lack of any effect in the field [157,158,159,160,161,162]. The P. *fluorescens* strains ACK55 and D7 inhibited the seed germination of B. *tectorum* in petri dish assays, but they showed no effects on B. *tectorum* during growth-chamber plant-soil assays and field tests [160,161]. The causes of this variation in the effect of the developed bioherbicides were not clear. The inconsistency in the herbicidal effects of weed-suppressive bacteria between in vitro screening and field application makes it difficult to develop bioherbicides from weed-suppressive PAB. Many factors should be considered for the successful development of PAB bioherbicides with a stable performance in the field.

### 8.1. The Host Range or Specificity

The host range of a weed-suppressive PAB must be studied before field assessments. Many bacterial pathogens not only infect unwanted weeds in the field, but also crops, especially crops in the same family as the targeted weeds. For example, *Pseudomonas* spp. and *Xanthomonas* spp. have a very wide range of hosts. P. *syringae* pv. *tagetis* is a common pathogen of sunflower, artichoke, and other crops. *Ralstonia solanacearum* is a common pathogen of Solanaceae plants, such as tomato, potato, pepper, and tobacco, etc. For the development of successful bioherbicides with bacterial pathogens of weeds, the host specificity is a key issue. Alongside the bacterial pathogens of weeds, all the other microbes must be evaluated for their host specificity. Microbes with a wide host range, especially those with pathogenicity to crops or desirable plants, must be avoided, or their application must be limited under certain ecological conditions.

Bacterial pathogens isolated from different weed hosts are indistinguishable by conventional biochemical tests, but they might show specificity to a certain host or to a broad spectrum of hosts. For example, when applied through spraying, the strains of P. *syringae* pv. *tagetis* isolated from common ragweed only infected ragweed, while other P. *syringae* pv. *tagetis* isolates from marigold, sunflowers, and Jerusalem artichoke caused leaf spots and apical chlorosis [53]. Different ecotypes of annual bluegrass (P. *annua*) responded differently to treatment with X. *campestris* pv. *poae* at a concentration of 10^9^ cfu mL^−1^. A single leaf-tip treatment controlled 80% of annual P. *annua* but only 60% of perennial P. *annua* [163]. This might have been caused by the rapid growth rate of perennial P. *annua*. In the screening for herbicidal DRB from crops and weeds, attention is always paid to the killing or growth inhibition effects in weeds. Some DRB also inhibit the growth of crops [164]. Although bacteria isolated from the roots of P. *hysterophorus* inhibited the root growth of P. *hysterophorus* in in vitro assays and under glasshouse conditions, they also decreased the root growth, root dry mass, and shoot length of crop plants [28]. Certain DRB introduced from the USA suppressed the growth of several weed species in Australia, but they also reduced the growth of some major Australian canola varieties [165]. The safety of herbicidal PAB for crops must be considered in bioherbicide development.

### 8.2. The Formulation of Herbicidal PAB

#### 8.2.1. Formulation Types

The formulation type affects the performance of herbicidal PAB in the field. A semolina–kaolin Pesta granules of P. *trivialis*X33d, showed greater suppression of the growth of B. *diandrus* and higher cell viability than talc–kaolin powder [80]. The formulation of P. *putida* and *Acidovorax delafieldii* with alginate maintained large populations of bacteria and produced a high concentration of cyanide in the soil, which reduced velvetleaf growth [166]. Water activity in the pesta granules significantly influenced the survival of P. *fluorescens* BRG100. Drying the pesta to 0.3 aw stabilized the population of BRG100 at 8.5 log10 cfu g^−1^ for 16 months. The starch types in the pesta formulation of P. *fluorescens* BRG100 also affected the disintegration of the pesta, with pea starch causing the fastest degradation [167]. The survival of P. *fluorescens* and its colonization of plants was improved by encapsulating the bacterium in alginated beads by incorporating skim milk or skim milk with betonite clay [168].

#### 8.2.2. Surfactants

Surfactants influence the infection of weeds by pathogenic bacteria. The nonionic organosilicone surfactant Silwet L-77 facilitated stomatal penetration by aqueous suspensions. The addition of 0.1% Silwet L-77 to a P. *syringae* pv. *tagetis* suspension increased the infection of Canada thistle and common ragweed by P. *syringae* pv. *tagetis*, causing 100% disease incidence and greater disease severity [169]. The addition of Silwet L-77 to a P. *syringae* pv. *phaseolicola* culture at 0.2% increased the infection of Kudzu leaves [170]. Silwet L-77 also greatly promoted the infection of *Xanthomonas* sp. LVA-987 [50]. The addition of Silwet L-77 (0.1% *v*/*v*) or Silwet 408 (0.2% *v*/*v*) to a P. *syringae* pv. *tagetis* (5 × 10^8^ cfu mL^−1^) suspension resulted in 100% disease incidence and greater disease severity in Canada thistle [52]. The addition of Silwet L-77 at 0.3% to P. *syringae* pv. *tagetis* increased the population of P. *syringae* pv. *tagetis* in Canada thistle leaves and significantly reduced the shoot dry weight [171].

#### 8.2.3. Carriers

The carriers in a formulation also affect the performance of PAB-based bioherbicides. Rakian et al. evaluated the effects of using talc, bentonit, kaolin, and ground burnt rice husk as carriers on the shelf life of two DRB, *Bacillus lentus* A05 and P. *aeruginosa* A08. They demonstrated that talc, bentonit, kaolin, and ground burned-rice husk were effective carriers for both bacteria, supporting populations of B. *lentus* A05 and P. *aeruginosa* A08 for up to 20 weeks and maintaining the stability of the two DRB’s biocontrol efficacy against A. *conyzoides* [172]. The inclusion of semolina flour in the formulation increased the effectiveness of P. *fluorescens* G2-11 in inhibiting the germination of seeds and the root growth of S. *viridis* [88]. A pesta formulation of P. *fluorescens* BRG100 incorporating oat flour and 20% maltose increased the survival of bacteria and prolonged the shelf life of the product from 3 weeks to 32 weeks [173]. The addition of soil and sucrose to a pesta granular formulation of P. *trivialis* X33d increased the stability of bacterial cells in the formulation [80].

### 8.3. Environmental Factors

Environmental factors such as the temperature, humidity, and soil properties affect the performance and survival of weed-suppressive PAB. Soil properties, such as the soil type, organic matter content, and pH, affect the performance of DRB in the suppression of weed growth. P. *putida* strain B1-7, P. *fluorescens* strain L2-19, and S. *maltophilia* strain TFR1 significantly inhibited green foxtail shoot growth in organically managed soil, which indicated that organic content is important for the growth-suppressive activity of DRB [87]. P. *fluorescens* G2-11 significantly inhibited the seed germination and plant growth of S. *viridis* in fine sandy loam [88]. Temperature is a key factor for the performance of herbicidal bacteria. Different weed-suppressive PAB demonstrate optimum performance at different temperatures, so the appropriate season or time of day for their application might vary. The optimal time for the application of P. *syringae* pv. *phaseolicola* to control kudzu might be late spring [52]. Mid-July was found to be the best time to apply P. *syringae* pv. *tagetis* to control Canada thistle for maximum disease incidence and disease severity, which might have been related to the precipitation levels during this time [59], and such control effects could be enhanced through consecutive applications. P. *fluorescens* strain D7 showed greater suppression of the growth of downy brome in cool or moist conditions, which favor the seed germination and establishment of downy brome [75]. The optimal time of day for the application of P. *syringae* pv. *tagetis* to control A. *grayi* was found to be noon [60]. The best time for the application of X. *campestris* pv. *poae* JT-P482 to control annual bluegrass was determined to be fall or winter; whereas treatment in October and February through to April produced acceptable levels of fresh-weight loss the following spring, and treatment in October, November, and February reduced seedheads by more than 75%, April treatment did not prevent the spread of seeds [44]. Temperatures of 20 °C/15 °C (day/night) enabled X. *campestris* pv. *poae* JT-P482 to effectively control annual bluegrass in glasshouse assays through heavy wilting or plant death [174].

### 8.4. Application Methods

The application method also affects the weed-suppressive performance of herbicidal bacteria. In a comparison of different application methods, P. *putida* strain FH160, S. *maltophilia* strain FH131, and E. *taylorae* strain FH650 showed a more consistent colonization of the rhizosphere of downy brome through soil incorporation [175], but the emergence of downy brome was reduced more substantially in the seed treatment via surface soil application compared to soil incorporation. Joint application with benlate, a systemic fungicide, could increase the populations of deleterious fluorescent pseudomonads in the rhizomes of leatherleaf fern and the expression of fern distortion syndrome [176]. When applied twice with a backpack sprayer, P. *syringae* pv. *tagetis* caused 71% apical chlorosis in Canada thistle, while this figure was only 18–51% for other application methods [171]. Backpack-sprayer application also caused a reduction in the height of shoots and the number of flower buds in Canada thistle, but the survival of Canada thistle was only reduced by 20%. Wounds were necessary for the effective control of annual bluegrass with X. *campestris* pv. *poae* JT-P482, and shortening the interval between cutting and the application of JT-P482 increased the efficacy [177]. The application of DRB with certain cover crops, such as *Brassica* and sweat clover, reduced the weed biomass by 90% in soybean fields and achieved a soybean yield that was higher than that of the weedy control and equivalent to or higher than that of conventionally grown soy [178]. In general, the seedlings of weeds are more sensitive to herbicides, no matter the pathogens or the microbes that have produced the herbicidal metabolites. The application of P. *syringae* pv. *tagetis* to small Canada thistle seedlings increased the efficacy of the treatment [58,59]. Mowing annual bluegrass increased the efficacy of P. *fluorescens* at limiting the spread of P. *annua* on football pitches [115].

The combination of certain simple chemicals with herbicidal PAB can potentiate their herbicidal activities. Although glycine alone had no effect on the growth of velvetleaf, when it was applied in combination with the cyanogenic rhizobacteria P. *fluorescens* and *Acidovorax delafieldii*, it increased the production of HCN in root-free soil [166]. The combination of 10^−5^ M L-tryptophan with E. *taylorae* increased the inhibition of the root growth of several species of weeds [99]. The addition of oil and sucrose, alone or in combination, to the formulation of P. *syringae* pv. *tabaci* used in biological weed control increased the survival of the bacteria; sucrose alone enhanced survival more than oil alone, but the beneficial effect of sucrose was reduced when it was combined with oil [179].

### 8.5. Endophytes in Weeds

Endophytes in weeds might counteract the efficacy of microbial weed-control agents. Some endophytic bacteria, such as *Bacillus* sp. isolates and *Ochrobactrum* quorumnocens RPTAtOch1, thwarted the control of angled onion with *Pectobacterium carotovorum* subsp. *carotovorum* because of its antagonism against P. *carotovoraum* subsp. *carotovorum* [180].

## 9. Conclusions and Prospects

In this review, we focused on herbicidal PAB, considering their screening, diversity, modes of action, and performance. PAB are widely present in the soil adjacent to plants and inside plants (weeds and crops). A rapid screening process is a prerequisite for selecting highly effective herbicidal bacteria from the broad spectrum of PAB. Some herbicidal PAB, such as X. *campestris* pv. *poae* and P. *fluorescens* strains D7 and ACK55, have been developed and registered as bioherbicides [15,16,17]. These demonstrate the potential of the development of bioherbicides using herbicidal PAB. Many PAB have shown seed germination or growth inhibition under in vitro screening, but their effects are reduced in pot assays and field experiments [157,158,159,160,161,162]. To ensure the consistent and stable performance of weed-suppressive PAB in the field, the formulation (including the types, surfactants, carriers, and other additives); application method (soil treatment, seed treatment, the wounding of weeds, sprayers, etc.); and environmental conditions (temperature, humidity, soil pH, and the content of organic matter in the soil) should all be considered and studied. Due to the existence of so many influencing factors, fluctuations in the performance of the weed-suppressive PAB in the field should be expected. The successful commercialization of certain PAB as bioherbicides has encouraged much effort to discover highly weed-suppressive PAB. Newly developed bioherbicides could be used as alternatives to conventional chemical herbicides or in cases where chemical herbicides are unsuitable.

In the development of bioherbicides based on PAB, we should consider their effects on crops and the environment and their safety for humans, as some herbicidal bacteria, such as K. *pneumoniae* and P. *aeruginosa*, are human pathogens. Although such human pathogens with herbicidal activities would not be approved as bioherbicides because of the strict registration regulations, it would be better to exclude them in the early stages of bioherbicide development. The application of herbicidal PAB with wide host ranges or crop pathogenicity, such as P. syringae and R. solanacearum, must be avoided or limited to specific areas to reduce their adverse effects on ecosystem balance and crop production. The application of herbicidal PAB for weed control might affect soil microbial communities, and more studies are required to clarify their effects on the entire soil microbiome of weeds and crops.

As the fields of microbiology, multi-omics, and next-generation sequencing progress, more PAB will be found. Abundant PAB from crop and weed microbiomes have been isolated and characterized according to their herbicidal activities, especially those from the rhizosphere [181]. As we know, culturable bacteria represent only a very small portion of the bacteria in plants and their adjacent environments. With metagenomic techniques, it is possible to uncover the phytotoxic or herbicidal allelochemicals produced by unculturable bacteria [182]. Field management to improve soil quality could change the microbiota in the soil, including weed-suppressive bacteria, which could promote natural weed suppression [183,184]. A more detailed study on plant–microbe interactions, including crop–microbe, weed–microbe, and crop–weed–microbe interactions, could help to elucidate the mechanisms of weed suppression and construct a simplified microbial consortium for weed control. We predict that further potential weed-suppressive PAB will be discovered and developed as bioherbicides.

## Figures and Tables

**Figure 1 plants-11-03404-f001:**
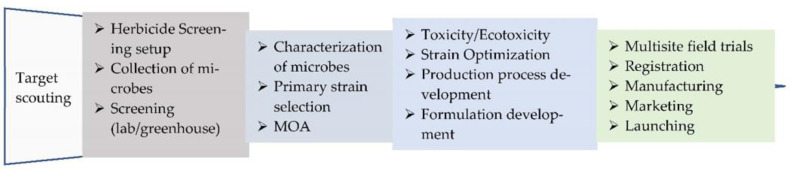
Diagram of steps involved in the development of bioherbicides based on microbes.

**Table 1 plants-11-03404-t001:** Bacterial pathogens of weeds with herbicidal activities.

Bacterial Pathogen	Target of Herbicidal Activities	Refs.
X. *campestris* pv. *poae*	Annual bluegrass (*Poa annua)*	[36,37,38,39,40,41,42,43,44,45]
X. *campestris*	Annual bluegrass, crabgrass, goosegrass, and other species of weeds	[46]
*Xanthomonas translucens* pv. *poae*	Annual bluegrass	[47,48]
X. *campestris* pathovar PT1	Poa trivialis	[49]
*Xanthomonas* sp. LVA987	*Xanthium strumarium*, *Conyza canadensis*, *Ambrosia artemisiifolia, Ambrosia trifida*, and some other Asteraceae plants	[50,51]
*Pseudomonas syringae* pv. *phaseolicola*	*Kudzu (Pueraria lobata)*	[52]
P. *syringae* pv. *tagetis*	Marigold, sunflower, Jerusalem artichoke, A. *artemisiifolia*, C. *canadensis*, *Lactuca serriola*, *Xanthium strumariaum,* and *Ambrosia grayi*	[53,54,55,56,57,58,59,60]
P. *syringae*-CT99B016C	*Cirsium arvense, Sonchus oleraceus*, *Sonchus asper (*annual and spiny sowthistle), and *Taraxacum officinale* (dandelion)	[61]
*Pseudomonas aeruginosa*	*Digitaria sanguinalis*	[62]
*Ralstonia solanacearum*	Kahili ginger (*Hedychium gardnerianum*)	[63]
*Burkholderia andropogonis*	Weeds in the Caryophllaceae, Poaceae, and Fabaceae families	[64]
*Bacillus cereus* XG1	D. *sanguinalis*	[65]

**Table 2 plants-11-03404-t002:** Deleterious rhizobacteria (DRB) with herbicidal activities.

Strain of DRB	Sources	Target Weeds	Refs.
P. *fluorescens* D7	The root of winter wheat	B. *tectorum*	[71,72,73,74,75,76]
P. *syringae* strain 3366	The rhizoplane of wheat	B. *tectorum*	[29]
*Pseudomonas putida* FH160, *Enterobacter taylorae* FH650, *Xanthomonas maltophila* FH131	The rhizoplane of downy brome	Downy brome, Japanese brome, and jointed goatgrass	[77,78]
*Pseudomonas trivialis* X33d	The rhizosphere of durum wheat	*Bromus diandrus*	[79,80]
P. *fluorescens* WH6	Rhizosphere soil	P. *annua*	[81]
P. *fluorescens* strains BRG100	The rhizosphere of green foxtail	Green foxtail	[82]
*Providentia rettgeri* strain CPS67, *Pseudomonas* isolate HWM11	The rhizosphere soil of wheat	*Phalaris minor*	[83]
Unidentified DRB strain Pk2	The rhizosphere of *Paspalum conjugatum*	P. *conjugatum*	[84]
*Pseudomonas kilonensis*/*brassicacearum* strain G11	*Galium mollugo*	*Echinochloa crus*-*galli*	[85]
*Chromobacterium* sp. S-4	The rhizosphere of D. *sanguinalis*	D. *sanguinalis*	[86]
Consortia of 3 *Pseudomonas* sp.	The rhizosphere of wheat	A. *fatua* and P. *minor* seedlings	[35]
*Bacillus* sp. X20	Wheat rhizosphere	A. *fatua*	[30]
P. *fluorescens* strain L2-19, *Stenotrophomonas maltophilia* strain TFR1, P. *putida* strain B1-7	The rhizosphere of green foxtail	Green foxtail	[87]
P. *fluorescens* G2-11	Roots of giant foxtail	Green foxtail	[88]
P. *fluorescens* strains XJ3, XS18, and LRS12	The soil and rhizoplane of plants in winter wheat fields	Annual bluegrass	[89]
P. *fluorescens* isolates LS102 and LS174	The rhizosphere of leafy spurge	Leafy spurge	[90,91]
*Flavobacterium balustinum* isolate LS105	The rhizosphere of leafy spurge	Leafy spurge	[90]
*Pseudomonas* sp. strain TR10	The rhizosphere of Palmer amaranth	*Amarathus palmeri*	[92]
P. *fluorescens* and *Acidovoras delafieldii*	The root surface and interior of *Abutilon theophrasti*	A. *theophrasti*	[93]
*Xanthomoas* sp, P. *aeruginosa*, P. *fluorescens*, *Bacillus subtilis,* and B. *cereus*	The rhizosphere of *Sida acuta*	S. *acuta*	[94]
P. *aeruginosa* isolate KC1	The rhizosphere of castor plants (*Ricinus communis*)	*Amaranthus spinosus* and *Portulaca oleracea*	[95]
DRB strain A08	The rhizosphere of *Ageratum conyzoides*	A. *conyzoides*	[84]
P. *aeruginosa* FS15	The soil adjacent to *Chenopodium* sp.	*Convovulus arvensis* and P. *oleracea*	[96]
*Pseudomonas asplenii* and P. *syringae*	The rhizoplane of white clover and ryegrass	White clover	[97]
P. *fluorescens* S611	roots of undetermined plants	White clover	[98]
E. *taylorae*	The rhizosphere of weeds	Several species of weeds	[99]
P. *fluorescens* WSM3455 and WSM3456 and *Alcaligenes xylosoxidans* WSM3457	The rhizosphere of wild radish	Wild radish	[100]
P. *putida* T42, P. *fluorescens* L9, P. *fluorescens* 7O_0_, P. *aeruginosa* O_0_10, and *Pseudomonas alcaligenes* W9	Weed-infested wheat soil	Broad-leaved dock	[101]
*Bacillus flexus* JMM24	The rhizosphere of *Lathyrus aphaca*	L. *aphaca*	[32]
*Bradyrhizobium japonicum* isolate GD3	Soybean roots	*Ipomoea hederacea*	[102]
P. *putida* GD4	Undetermined source	I. *hederacea*	[102]
P. *fluorescens* strain QUBC3	The root of *Phalaris* sp.	*Orobanche aegyptiaca* and *Orobanche cernua*	[27]
P. *fluorescens*/P. *putida* isolates	Soils naturally supressive to *Striga hermonthica*	S. *hermonthica*	[103]

## Data Availability

Not applicable.

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
