# Peer review of "Plant-Associated Bacteria as Sources for the Development of Bioherbicides"

_plants, 2022, doi:10.3390/plants11233404_

Round 1

Reviewer 1 Report

Fang et al summarize current advances about bacteria-related bioherbicides, the review provides many different cases about bioherbicides. I would like invite the authors to integrate the whole story to make it easier to read.

1. Charts are needed to show the bioherbicides from different bacteria for different weeds.

2. Please draw the diagrams to show the steps to develop bioherbicides.

Author Response

Thanks a lot for the comments and suggestions on the manuscript. Please find following the responses. The manuscript was also revised carefully as showed in the revised version.

Point 1. Charts are needed to show the bioherbicides from different bacteria for different weeds.

 Response 1: As suggested by the reviewer, the different plant-associated bacteria with herbicidal activities had been summarized in 2 tables, with one about the bacterial pathogens of weeds, the other one about the deleterious rhizobacteria. Regarding other types of plant-associated bacteria, because of the limited number of the bacteria, readers can easily get the info of those bacteria, I thought that it is not necessary to summarized them in separate tables.

 Point 2: Please draw the diagrams to show the steps to develop bioherbicides.

Response 2: The steps to develop microbe-based bioherbicides as drown as following. The figure was inserted in the manuscript in line between 69 and 70.

Reviewer 2 Report

Comments to manuscript „Plant-associated bacteria as the sources for the development of bioherbicides“ submitted by Fang  et al.

The manuscript presents a spanning collection of available methods for the screening of herbicidal bacteria followed by a compilation of bacteria with possible and known functions eventually usefull for weed control, such as pathogens, herbicial rhizobacteria, PGPR bacteria with inhibitory effects on parasitic Orobanche, and many others, including endophytes.The mode of action is addressed and some important bacterial phytotoxins are mentioned. Also few limitations and constraints of the use of those bacteria are presented. The authors reviewed important recent and older articles dealing with the topic. The relevant literature is presented.

My major concern is the rather uncritical handling of the use bacteria for weed control in agriculture. For instance, it is not mentioned that Klebsiella pneumoniae or Pseudomonas aeruginosa are redoubtable human pathogens, other bacteria are pathogens for wild plants, but also most of the other bacteria presented in the manuscript can have effects on wild plants, as bacteria can escape from the arable soil. Natural plant communities can be severely damaged. The effects on soil microbial communities and on plant microbiomes can be dramatic, when additional bacteria get in conflict with species already colonizing the plant. It is also not mentioned that microbial phytotoxins and allelochemicals can alter the entire soil microbiome with negative effects on the ecosystem by eliminating sensitive members of the communities. On the other hand, evolutionary processes,  which may be disadvantageous  for agriculture, can be initiated due to the imbalanced communities.

It is recommended to include a paragraph where these severe problems are discussed, particularly in our era of dramatic loss of species diversity, due to human activities.

Because of the many type and syntax errors it is necessary to correct the English by a Native speaker. All scientific plant names should in italics.

Author Response

Thanks for the good comments on the manuscript. The manuscript was revised as suggested with an additional paragraph regarding the problems, and the syntax and typing errors had been carefully checked and corrected. 

The manuscript was also revised thoroughly, with some paragraphs and the sequence of reference reorganized.

Best wishes

Kaimei

Round 2

Reviewer 1 Report

much better

Reviewer 2 Report

Dear Authors.

A second revision is not necessary.